# Gut microbiota *Turicibacter* strains differentially modify bile acids and host lipids

Jonathan B. Lynch [1,6] ✉, Erika L. Gonzalez[1], Kayli Choy[1], Kym F. Faull[2,3,4], Talia Jewell[5], Abelardo Arellano[5], Jennifer Liang[5], Kristie B. Yu[1], Jorge Paramo[1] & Elaine Y. Hsiao [1]

Bacteria from the *Turicibacter* genus are prominent members of the mammalian gut microbiota and correlate with alterations in dietary fat and body weight, but the specific connections between these symbionts and host physiology are poorly understood. To address this knowledge gap, we characterize a diverse set of mouse- and human-derived *Turicibacter* isolates, and find they group into clades that differ in their transformations of specific bile acids. We identify *Turicibacter* bile salt hydrolases that confer strain-specific differences in bile deconjugation. Using male and female gnotobiotic mice, we find colonization with individual *Turicibacter* strains leads to changes in host bile acid profiles, generally aligning with those produced in vitro. Further, colonizing mice with another bacterium exogenously expressing bile-modifying genes from *Turicibacter* strains decreases serum cholesterol, triglycerides, and adipose tissue mass. This identifies genes that enable *Turicibacter* strains to modify host bile acids and lipid metabolism, and positions *Turicibacter* bacteria as modulators of host fat biology.

The gut microbiota forms complex relationships with its host organism, modulating broad aspects of host physiology including metabolism[1,2] and neurobiology[3,4]. Often, the connections between the gut microbiota and host physiology are easiest to decipher through presence/absence of large sectors of the microbial community (examples in refs. [5–7]), but in some cases, specific microbial features and/or taxa serve important roles in host biology[8–10].

The bacterium *Turicibacter sanguinis* is the best-studied species of the *Turicibacter* genus, which is prevalent in the gut microbiota of several animals, including humans[11], mice[7], cows[12], pigs[13], and chickens[14]. *Turicibacter* can reach relative abundances of >20% in the rodent small intestine[15] and 0.5% in the human fecal microbiota[16,17]. A study of 1126 sets of twins found *Turicibacter* was a highly heritable

taxon in the human fecal microbiota[18], supporting associations between *Turicibacter* abundance and host genetics found in Diversity Outbred mouse collections[19–21]. Long-standing co-evolutionary connections between this taxon and humans are suggested by the high abundance of *Turicibacter* sequences that have been measured in ancient and non-industrial human gut microbiotas[17,22–24].

The mammalian gut microbiota has long been associated with obesity[25,26], but studies often provide strong correlations rather than mechanistic determinants of these relationships, indicating a further need for fundamental interrogation into connections between the microbiota and host fat[27]. Numerous microbiota community profiling studies reveal correlations between *Turicibacter* and features of host fat metabolism, such as adiposity

[1]Department of Integrative Biology & Physiology, University of California, Los Angeles, Los Angeles, CA 90095, USA. [2]Department of Psychiatry and Biobehavioral Sciences, University of California, Los Angeles, Los Angeles, CA 90095, USA. [3]Jane and Terry Semel Institute for Neuroscience and Human Behavior, University of California, Los Angeles, Los Angeles, CA 90095, USA. [4]Pasarow Mass Spectrometry Laboratory, University of California, Los Angeles, Los Angeles, CA 90095, USA. [5]Isolation Bio, San Carlos, CA 94070, USA. [6]Present address: Department of Biological Chemistry, Johns Hopkins University School of Medicine, Baltimore, MD 21205, USA. ✉e-mail: jlynch48@jhmi.edu

and dietary lipids[28–33], but the nature of these correlations varies[34,35]. We recently observed that the type strain of *T. sanguinis*, MOL361[36,37], broadly alters the host serum lipidome while decreasing serum triglycerides in mice[38]. This same strain was also reported to modify bile species through deconjugation and dehydrogenation in vitro[21], suggesting at least one potential means by which *Turicibacter* can influence host lipid status. Based on these findings, we hypothesized that there may be variations in the functional activity of *Turicibacter* strains that account for differences in host bile and lipid biology, providing a mechanism to connect this taxon to aspects of host physiology.

In this work, we show that diverse *Turicibacter* isolates influence host metabolites, including lipids and bile acids. These isolates each perform bile transformations in vitro, albeit with differing chemical specificity. We identify and describe bile modifying genes from these isolates and measure their effects on host physiology. We find that these genes are sufficient to broadly alter host lipid and cholesterol states, presenting a means by which bacteria from the *Turicibacter* genus may modulate biology of their diverse hosts.

## Results

### *Turicibacter* isolates separate into genetically distinguishable strains

To better understand the diversity within the *Turicibacter* genus, we gathered nine isolates from the fecal microbiotas of mice and humans that had been identified as *T. sanguinis* based on their 16S rRNA gene sequence (97% full length 16S rRNA gene sequence similarity cutoff, Supplementary Data 4). Two of these isolates had

been previously identified (human-derived type strain MOL361 and H121, which was derived from contaminated germ-free mice[36,39]); five had been isolated but not published (human isolates 18F6, T46, and T129, and mouse isolates 1E2 and TA25); and two were isolated from a human fecal sample specifically for this study (GALT-E2 and GALT-G1) using an array-based isolation and cultivation platform (see Methods). We performed shotgun short read sequencing and created draft assemblies of each isolate genome. Comparisons of the 16S rRNA gene phylogeny (Fig. 1a), general genome characteristics (Fig. 1b), or specific genome sequence (Fig. 1c, d) revealed that even with this fairly small sample of the 16S rRNA gene-based "*T. sanguinis*" species designation, there were at least three distinct subgroups: two from humans (exemplar isolates MOL361 and H121, with 99.3% full length 16S rRNA gene similarity), and one from mice (exemplar strain 1E2, 97.5% and 97.8% 16S rRNA gene similarity with MOL361 and H121, respectively). Genomic alignments indicated a substantial amount of shared DNA sequences within members of the same subgroups (all within group average nucleotide identity [ANI] >98.3%) with the remaining amount of genetic variation indicating smaller genetic differences between related isolates. These within-subgroup shared sequences were distinct from members of the other two subgroups (intergroup ANI: MOL361-H121 = 76.80%, MOL361-1E2 = 74.95%, H121-1E2 = 77.43%). It is important to note that the H121-group genomically resembles the newly described species *Turicibacter bilis*[13] (98.8% ANI), currently the only other named species from this genus. Overall, these genomic differences suggest distinct evolutionary histories that correspond at least partially with host origin.

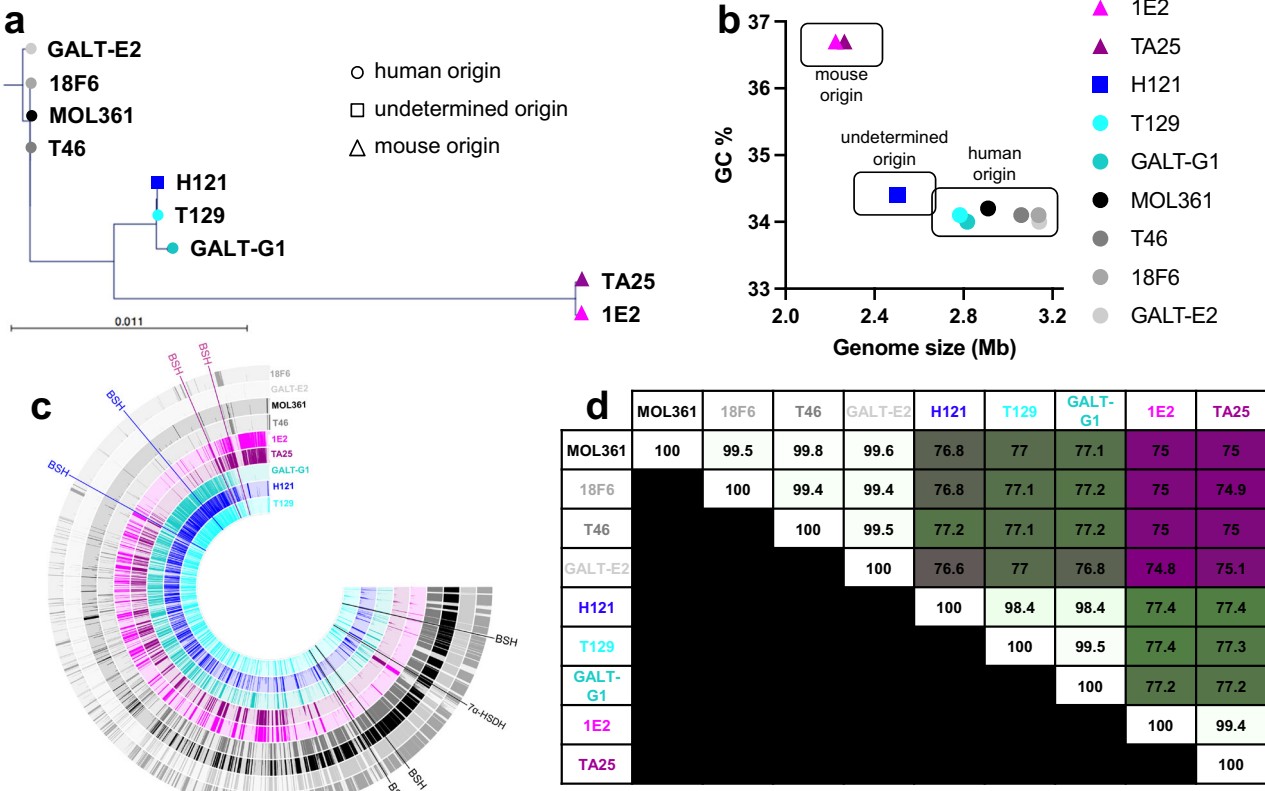

**Fig. 1 | Genomic comparisons reveal distinct subgroups of *Turicibacter*.**
**a** Phylogenetic tree comparing full-length 16S rRNA gene sequences from noted *Turicibacter* isolates. Circles indicate human-derived isolates, triangles indicate mouse-derived isolates, and square indicates a mouse-derived contaminating isolate. **b** Association between guanine-cytosine % (GC%) and calculated genome size in megabases (Mb) for shotgun-assembled genomes of *Turicibacter* isolates from **a**. **c** Full genome sequence comparisons across *Turicibacter* strains. Position of predicted bile-modification gene homologs are noted outside of rings, with the color of the gene name denoting the genome family that gene is found in. Each ring represents sequence blocks in one genome. **d** Average nucleotide identity (ANI) between noted *Turicibacter* genomes. Number denotes ANI, white-green-maroon scale represents 100%-75% ANI scale.

### *Turicibacter* isolates differ in their impact on host fat biology and circulating metabolome

Previous findings revealed that monocolonizing mice with *T. sanguinis* MOL361 altered host fat tissue and circulating lipids[38]. Due to the large genomic variation between our *Turicibacter* strains, we predicted that they would vary in their effects on host lipid biology. We chose representative isolates from each of the distinct phylogenetic sub-groups (MOL361, H121 and T129, and 1E2) and measured their effects on circulating metabolites and adipose tissue in monocolonized mice relative to germ-free (GF) and conventionalized controls (*i.e.*, gavaged with complete microbiota, CONV). Compared to GF mice, CONV mice had decreased levels of several dicarboxylate fatty acids, long-chain fatty acids, and long-chain acyl carnitines, with a broad increase in short- and medium-chain acyl carnitines (Fig. 2a). Consistent with a previous report[38], colonization with individual *Turicibacter* strains also induced widespread alterations in host serum lipids, with several strain-level differences in host lipid alterations (Fig. 2a, Supplementary Data 5). Compared to GF controls, MOL361 increased a subset of long-chain acyl carnitines, and decreased many long-chain saturated fatty acids. In addition, colonization with MOL361 elicited trending decreases in host cholesterol to levels below those seen in GF (Fig. 2b). Compared to GF, H121 colonization significantly increased serum levels of several medium-chain fatty acids, dicarboxylic acids, and short-, medium- and long-chain acyl carnitines. 1E2 colonization had a smaller overall effect on host lipids, but led to a decrease in several dicarboxlyate fatty acids (Fig. 2a). In addition to differences between GF or CONV mice and *Turicibacter*-monocolonized animals, there were also alterations in host lipids that showed some strain-dependency, with trending discrepancies in dicarboxylate fatty acids and significant differences in long chain saturated fatty acids (Fig. 2a) At the tissue level, two of the four *Turicibacter* strains stimulated statistically significant increases in epididymal/gonadal white adipose tissue (e/gWAT) mass in comparison to GF controls, and a third strain elicited similar increases that were not statistically significant (T129). In contrast, there was no noticeable effect of H121 on e/gWAT mass (Fig. 2c, Supplementary Data 6), distinguishing it from the other *Turicibacter* strains. Consistent with this, H121 showed the smallest e/gWAT adipocyte size within fat pads (Supplementary Fig. 1a–f). This may be due to lower colonization of H121 in both the small intestine and the colon (Supplementary Fig. 1g, h) The effects of particular *Turicibacter* strains on serum lipids, cholesterol, and fat mass directed us to investigate molecules that could potentially connect these biomolecules: bile acids.

Bile acids can affect circulating host lipids by altering fat digestion and systemic hormonal signaling[40]. They are produced by the host and released into the small intestine, where they promote digestion of fats by facilitating micelle formation, and can also act through receptors like the farnesoid X receptor (FXR)[40] and GPBAR1/TGR5[41]. Gut bacteria can modify bile acids, primarily through transformations like deconjugation. Previous reports found that MOL361 can broadly modify bile acids in vitro[21], so to determine if these abilities allowed MOL361 and other *Turicibacter* strains to modify host bile acids in vivo, we profiled serum and cecal bile acids in *Turicibacter*-monocolonized mice. Though each strain had some unique impacts on host serum metabolites, we noted consistent patterns in bile acids across monocolonized mice in comparison to either their GF or CONV controls. (Note: bile species with amino acid conjugants are typically referred to as "bile salts," but for simplicity, we will herein refer to both conjugated and unconjugated bile species as "bile acids"). Colonization with all of the *Turicibacter* strains led to a general increase in serum levels of unconjugated primary bile acids like cholate (CA), chenodeoxycholate (CDCA), and β-muricholate (βMCA) (Fig. 3a–d), and a similar rise in unconjugated secondary bile acids 3-dehydrocholate and 7-ketodeoxycholate (Fig. 3e, f). These responses were highly variable in the case of T129 colonization, leading us to de-emphasize this strain for

subsequent experiments. In response to colonization with MOL361, 1E2, or H121, we also found a similar increase in several unconjugated bile acids in the cecum (Supplementary Fig. S2). In all, these results suggest that these *Turicibacter* strains are able to impact host bile acids, potentially by deconjugating them in the gut.

When we investigated potential explanations for increases in unconjugated bile species, we noticed that the serum levels of conjugated bile acids differed between animals colonized by different *Turicibacter* strains, with the clearest delineation separating H121 from MOL361 and 1E2. MOL361- and 1E2-colonized animals generally had lower levels of taurine-conjugated primary bile acids in comparison to H121-colonized animals (Fig. 3g–j), whereas H121 colonization led to an increase in glyco-beta-muricholic acid (GβMCA) (Fig. 3k). Female mice displayed the most significant *Turicibacter*-associated changes in circulating bile species, lipids, and cholesterol, indicating sex-dependent responses to *Turicibacter* colonization (Supplementary Fig. 3). This sex difference has also been reported in C57BL/6 mice monocolonized with *T. sanguinis* MOL361, albeit with a directionally different change in adipocyte size[38]. Notably, in the cecum, there was less strain-specific distinction in conjugated bile acids, with taurine-conjugated bile acids decreased by colonization with any of the three *Turicibacter* strains we surveyed (Supplementary Fig. S2). The one cecal bile acid that did display strain dependency was the secondary bile acid lithocholic acid (Supplementary Fig. S2, bottom). Also, the low levels of detected glycine-conjugated bile acids (specifically glyco-cholic acid) in both serum and cecal contents was slightly increased by *Turicibacter* colonization (Fig. 3k, l; Supplementary Fig. S2c) Overall, these data demonstrate that colonization with *Turicibacter* alters serum bile acids, lipids, and cholesterol, as well as host fat mass. Further, while some changes like increased unconjugated bile acids were conserved across *Turicibacter* colonizations, levels of specific conjugated bile acids differed between *Turicibacter* strains, with MOL361 and 1E2 leading to lower levels of taurine-conjugated bile acids than H121 colonization.

### *Turicibacter* strains differ in their ability to modify host bile acids

Based on the large genetic variation between the *Turicibacter* strains (Fig. 1) and the differences in aspects of serum lipid and bile acid profiles seen in response to colonization with different isolates (Fig. 3), we posited that Turicibacter strains differ in their ability to modify bile acids. To test this, we grew each of our nine isolates to stationary phase in rich medium supplemented with a sub-inhibitory concentration of five bile species[21]: cholic acid, chenodeoxycholic acid, deoxycholic acid (DCA), taurocholic acid (TCA) and glycochenodeoxycholic acid (GCDCA). We then used liquid chromatography-mass spectrometry (LC-MS) to characterize the resulting bile transformations performed by each isolate (Fig. 4a). We discovered that the strains not only differed in their ability to modify this combination of bile species, but also that these capabilities generally mirrored the groupings identified in genomic comparisons (Fig. 4b). MOL361, 18F6, and GALT-E2 deconjugated both tauro- and glyco-bile acids, and also dehydrogenated CA and CDCA (Fig. 4b). 1E2 and TA25 deconjugated tauro-bile acids, but did not detectably deconjugate glyco-bile acids or perform dehydrogenation (Fig. 4b). H121 and T129 deconjugated glyco-bile acids, but did not readily deconjugate tauro-bile acids nor perform detectable dehydrogenation (Fig. 4b). T46 and GALT-G1 did not have bile-modifying capacity that mirrored their genetic phylogeny; T46 genomically resembled the MOL361 group but performed modifications similar to the H121 group (*i.e.* glyco- but not tauro- deconjugation, minimal dehydrogenation), whereas GALT-G1, which genomically resembled the H121 group, performed more MOL361-like transformations (*i.e.*, glyco- and tauro-deconjugation, dehydrogenation of CDCA) (Figs. 1, 4b, Supplementary Data. 1). Overall, each strain performed at

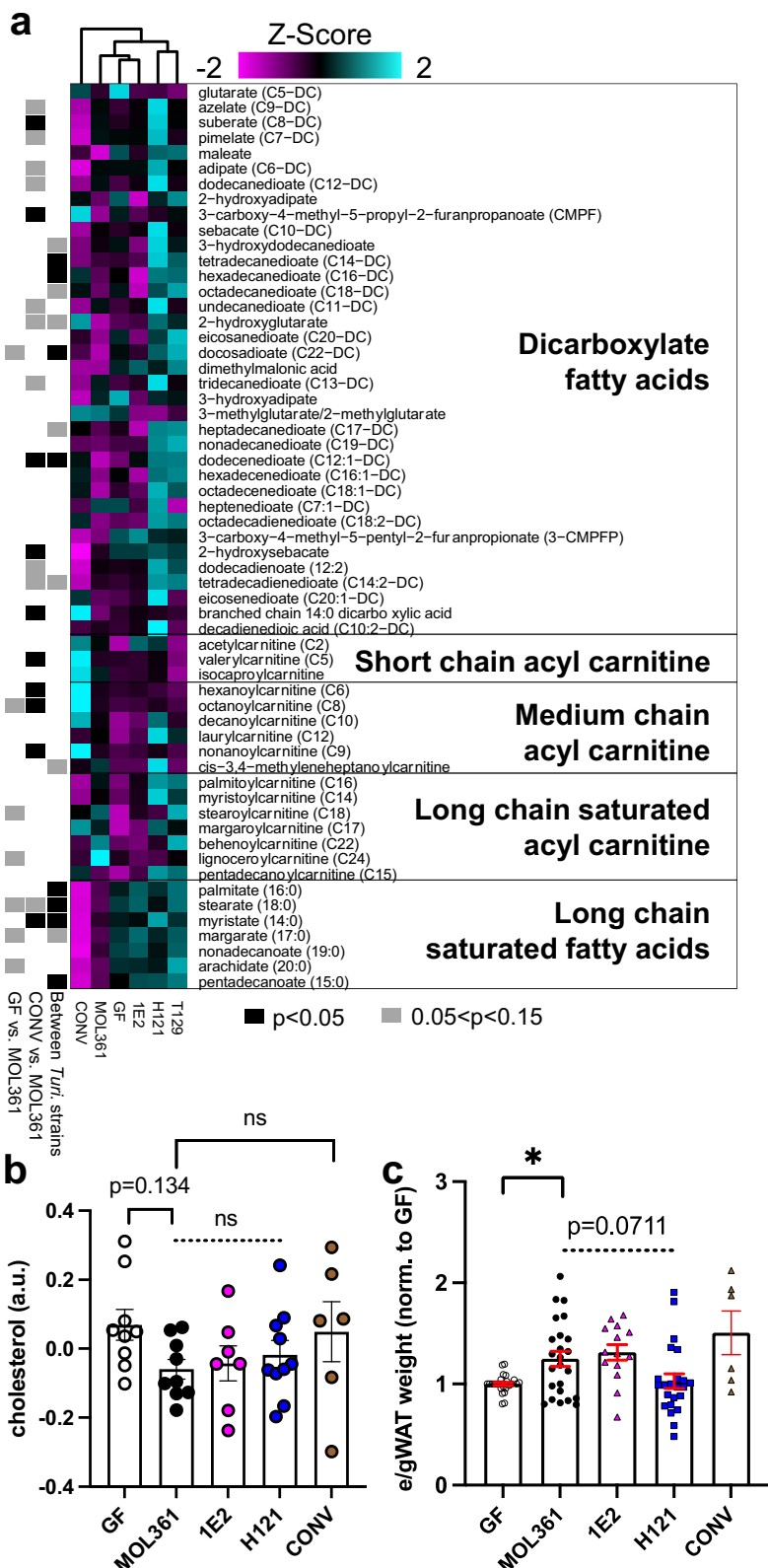

least one of three bile transformations, with some showing capacity for all three (Fig. 4b, Supplementary Data 1).

To confirm these intragroup distinctions, we chose one isolate from each of the subgroups (MOL361, H121, 1E2), and grew them in the presence of four primary conjugated bile acids: TCA, taurochenodeoxycholic acid (TCDCA), glycocholic acid (GCA), and GCDCA. This supported the same pattern seen above; MOL361

deconjugated both groups of bile acids, 1E2 preferentially deconjugated tauro-conjugates, and H121 preferentially deconjugated glyco-conjugates (Fig. 4c). MOL361 and 1E2 displayed broad deconjugation of tauro-conjugates and were able to process at least six taurine-conjugated bile acids (Supplementary Fig. 4). These data reveal that while all tested strains are efficient modifiers of bile species, their specific transformations differ in a strain-dependent manner,

**Fig. 2 | Colonization with different *Turicibacter* strains alters host lipids.**
**a** Heatmap displaying relative abundance of serum lipids from gnotobiotic mice monocolonized with noted *Turicibacter* strains. Heatmap values (Z-score) represent mean abundance of each detected lipid species from labeled lipid categories scaled across all the means of that individual lipid species. Black ($p < 0.05$) and gray ($0.05 < p < 0.15$) rectangles indicate statistically significant differences of that metabolite between: left, GF and MOL361 monocolonized mice; middle, CONV and MOL361 monocolonized mice; and right, between MOL361, 1E2, and H121 monocolonized mice. **b** Serum cholesterol abundances of mice colonized by noted *Turicibacter* strains. **c** Sex and litter-matched relative epidydimal/gonadal white

adipose tissue (e/g WAT) mass of mice monocolonized with noted *Turicibacter* strains. Shapes indicate value for individual mouse, dotted bar represents ANOVA statistic for groups below the line, a.u. = arbitrary units. WAT analysis $n$: GF = 26, MOL361 = 24, 1E2 = 14, H121 = 23, T129 = 10, CONV = 6. All other analysis $n$: GF = 9, MOL361 = 9, 1E2 = 7, H121 = 10, T129 = 7, CONV = 6. In all panels Mann–Whitney test for MOL361-GF and MOL361-CONV comparisons, Kruskal–Wallis for intra-*Turicibacter* comparison, Šidák correction for multiple comparisons[86]. Errors bars are mean +/− SEM, *$p < 0.05$, GF-MOL361 in 2c: $p = 0.0108$. Data are provided as source data file.

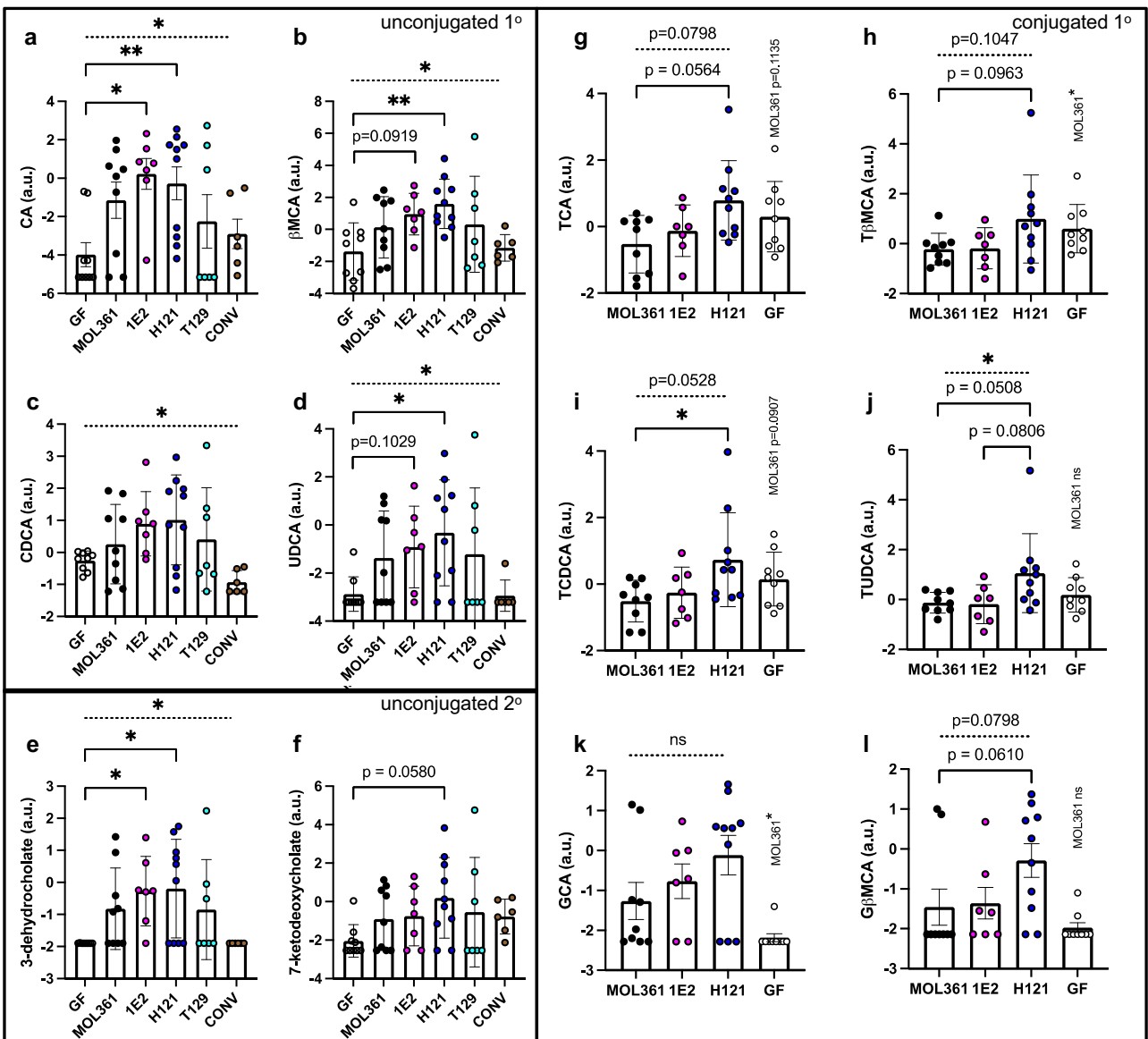

**Fig. 3 | *Turicibacter* colonization alters circulating host bile species in a strain-specific manner.** Serum abundances of **a**–**d**, primary unconjugated bile acids; **e**, **f** secondary unconjugated bile acids; or **g**–**l** primary conjugated bile acids. Points indicate log-transformed value for individual mouse with shapes and colors matching Fig. 1, error bars represent mean +/− SEM, a.u.=arbitrary units. Kruskal–Wallis test across all noted colonizations with Dunn's multiple comparisons to GF for **a**–**f**. Kruskal–Wallis test between noted *Turicibacter* strains and multiple comparisons to H121 for **g**–**l**, and *p*-values are noted above GF data points. Mann–Whitney test used to compare GF and MOL361 in **g**–**l**, and *p*-values are noted above GF data points. $n$ for each group: GF = 9, MOL361 = 9, 1E2 = 7, H121 = 10, T129 = 7, CONV = 6. Dotted bar represents

ANOVA statistic for groups below the line. Error bars are mean +/− SEM, *$p < 0.05$, **$p < 0.005$. Corrected *p*-values (ANOVA/GF-MOL361/GF-1E2/GF-H121/GF-T129/GF-CONV): 2a = (0.0204/0.1785/0.0141/0.0049/0.999/0.999); 2b = (0.0116/0.5123/0.0919/0.0059/0.9256/0.999); 2c = (0.0208/0.999/0.5425/0.5306/0.999/0.4158); 2d = (0.0216/0.4850/0.1029/0.0201/0.8856/0.999); 2e = (0.0121/0.3585/0.0299/0.0405/0.7612/0.999); 2f = (0.2203/0.6556/0.6645/0.0580/0.999/0.4012). Corrected *p*-values (ANOVA/H121-MOL361/H121-1E2/GF-MOL361): 2g = (0.0798/0.0564/0.3171/0.1135); 2h = (0.1047/0.0963/0.2315/0.400); 2i = (0.0528/0.0351/0.2726/0.0907); 2j = (0.0407/0.0508/0.0806/0.2973); 2k = (0.3988/0.3757/0.7675/0.0226); 2l = (0.0769/0.0610/0.2393/0.7176).

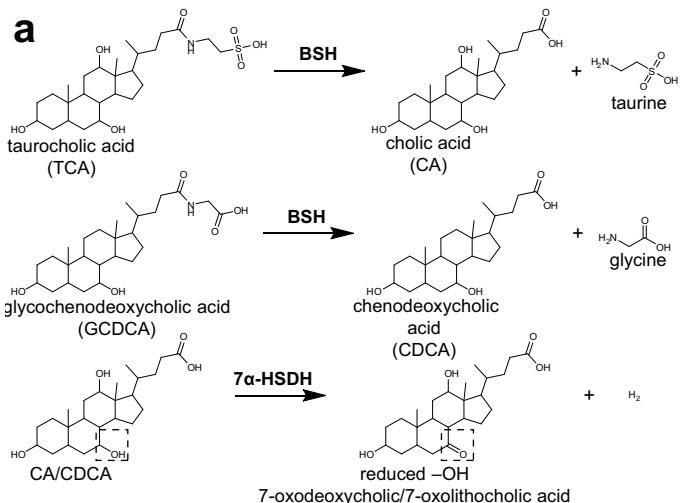

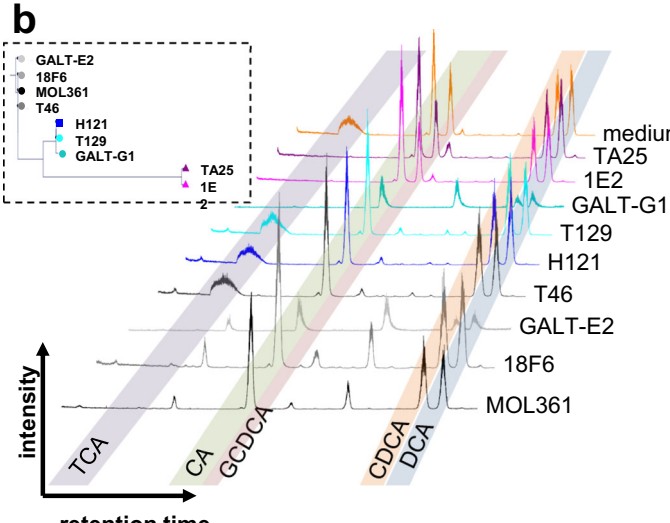

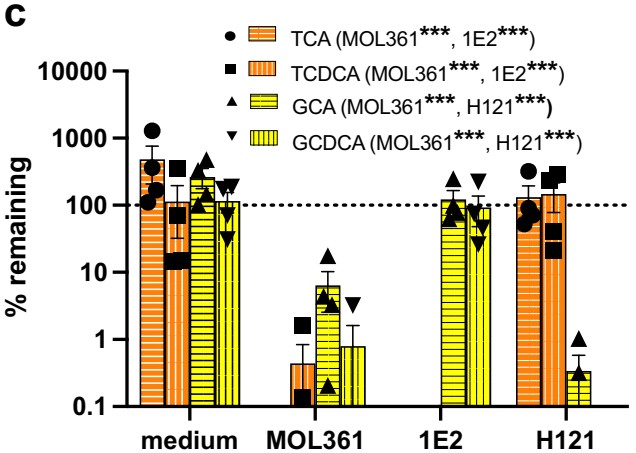

**Fig. 4 | *Turicibacter* isolates differ in their bile-modifying abilities. a** Schematic for types of bile transformations found to be performed by *Turicibacter* isolates. **b** inset: 16S rRNA gene-based phylogenic tree from Fig. 1a. Liquid chromatograms of individual *Turicibacter* isolates grown for 24 h in media with sub-inhibitory concentrations of five bile acids: taurocholic acid (TCA), cholic acid (CA), gly-cochenodeoxycholic acid (GCDCA), chenodeoxycholic acid (CDCA), and deoxy-cholic acid (DCA). Shaded regions indicate expected retention time of each bile species. **c** Percent remaining (compared to cultures at time = 0) of conjugated bile acids (TCA, taurochenodeoxycholic acid [TCDCA], glycocholic acid [GCA], GCDCA) after 24 growth with noted *Turicibacter* isolate. Yellow = glycine-conjugated bile acids, orange=taurine-conjugated bile acids. $n$ = 4 independent cultures. Values not shown were below 0.1% remaining. Statistical analysis performed by one sample $t$-test, annotations of legend denotes strains with significant difference from 100% remaining for each bile acid. $p$-values for each bile acid (MOL361/1E2/H121): TCA = (< 0.0001/ < 0.0001/0.6327); TCDCA = (< 0.0001/ < 0.0001/0.5410); GCA = (0.0002/0.6348/ < 0.0001); GCDCA = (< 0.0001/0.8803/ < 0.0001). Data are pro-vided as source data file.

genes. Certain bacteria from the gut microbiota dehydrogenate hydroxyl groups from the steroid core of bile acids[42], increasing their polarity and modulating their affinity for host bile acid receptors[43]. We searched the *Turicibacter* genomes for homologs of the characterized 7α-hydroxysteroid dehydrogenase (7α-HSDH[21]) from *Clostridium absolum*[44]. This revealed genes with 57% amino acid identity in MOL361, 18F6, T46, and GALT-E2, and homologs with 59% amino acid identity in H121 and T129 (Supplementary Fig. 5a). Though the H121-derived putative homolog had higher overall sequence identity than the MOL361-derived homolog, it lacked certain features predicted to be critical for dehydrogenase activity, such as the analogous Asp38 that is catalytically critical for this reaction[44]. Because in vitro experiments showed only isolates from the MOL361 group performed bile dehydrogenation, we cloned the putative 7α-HSDH homolog from MOL361 into *E. coli* C41-pLys and then grew these cells in individual unconjugated bile acids that can be dehydrogenated: CA, CDCA, and DCA. The protein encoded by the gene removed the mass equivalent of two hydrogens from CA (Supplementary Fig. 5b, c). Background transformation by *E. coli* prevented clear evidence of CDCA dehy-drogenation by this putative 7α-HSDH homolog (Supplementary Fig. 5d), but this homolog did not act on DCA (Supplementary Fig. 5e), supporting its annotation as a 7α-HSDH.

Conjugation increases bile acid solubility and emulsification ability[45], while deconjugation reverses these effects, leading to decreased dietary lipid absorption. To identify *Turicibacter* bile salt hydrolase (*bsh*) genes responsible for the strain-specific differences in bile deconjugation, we first searched our assembled genomes for annotations of "choloylglycine hydrolase," the broad category that includes these genes. Of these annotated genes, we identified eight groupings of homologous sequences, and found that each *Tur-icibacter* strain encodes putative choloylglycine hydrolases from at least two of the eight groups (Fig. 5a, b). Isolates within the same phylogenetic and phenotypic subgroups largely shared similar sequences (Fig. 5a, b). To assay the function of the strain-specific putative *bsh* genes, we cloned and individually expressed one representative sequence from each of the eight putative cho-loylglycine hydrolase groups in *E. coli* C41-pLys and measured the ability of these engineered bacteria to perform the deconjugations we observed in our native *Turicibacter*. We cultured the individual *E. coli* strains to stationary phase in the presence of two tauro- (TCA, TCDCA) or glyco- (GCA, GCDCA) bile acids, then measured their ability to deconjugate these bile acid pools. We found that *E. coli* expressing four of the eight putative *bsh* gene groups showed deconjugation activity against at least one of the bile acids (Fig. 5c–e). From MOL361, one BSH (group IV) is tauro-specific, and one (group I) deconjugates both glyco- and tauro-conjugates (Fig. 5c, d). 1E2 shares a tauro-specific BSH with MOL361 (group

potentially reflecting functions that influence their differential effects on host lipid biology.

**Turicibacter genomes have different repertoires of bile salt hydrolases**

The different bile modification abilities across strain subgroups sug-gested that each contained unique repertoires of bile-modifying

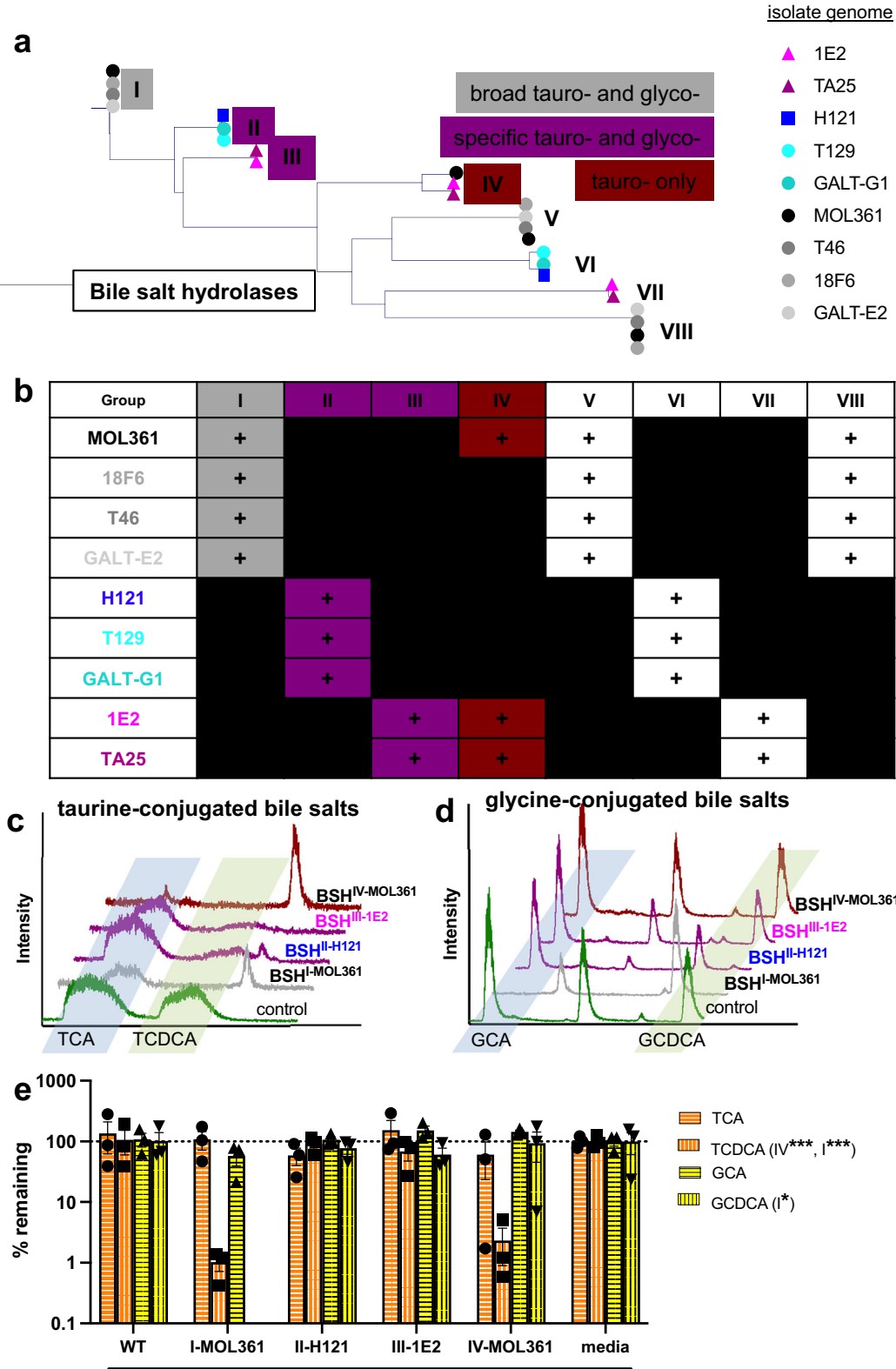

IV), and has another BSH (group III) with moderate activity on TCDCA (Fig. 5c, d). H121 has a BSH (group II) with activity on TCDCA and GCDCA (Fig. 5c, d), although this was blunted when presented with the combined four bile acids (Fig. 5e), potentially because of competitive inhibition. Collectively, these findings reveal that *Turicibacter* strains contain a range of bile salt hydrolases with deconjugation preferences for different bile acids.

**Strain- and substrate-specific bile salt hydrolases from *Turicibacter* differentially alter host lipid composition**

Given that *Turicibacter* colonization broadly modified host lipid and bile pools (Fig. 2), and that bile transformations have been previously shown to alter host lipids[46,47], we predicted that expressing *Turicibacter* bile-modifying genes outside the context of *Turicibacter* colonization would be sufficient to impact host lipid biology. To

**Fig. 5 | _Turicibacter_ isolates differ in their genetic capacity to modify bile species. a** Phylogenetic tree of amino acid sequences for each predicted bile salt hydrolase (BSH) sequence from _Turicibacter_ isolates, with observed bile species specificity noted in boxes. We did not detect bile salt hydrolase activity in sequences without boxes, representing groups V-VIII. **b** Presence (+) or absence of sequence homologs with potential BSH activity in _Turicibacter_ isolates. **c** Liquid chromatograms of media after 24 h cultures of _E. coli_ expressing individual predicted _bsh_ genes from each sequence grouping and grown with TCA and TCDCA. Control is _E. coli_ with same expression vector but expressing non-bile-modifying gene. **d** same as **c** but with GCA and GCDCA instead of tauro-bile acids.

**e** Quantification of percent remaining (compared to media controls) of conjugated bile acid (TCA, TCDCA, GCA, GCDCA) after 24 h growth with _E. coli_ expressing the noted _Turicibacter bsh_ gene. $n = 3$ independent cultures, annotations in legend indicate *$p < 0.05$, **$p < 0.005$, ***$p < 0.0005$ using one sample _t_-test comparison with 100% remaining. _p_-values for each bile acid expressed in _E. coli_ (WT/BSHI-MOL361/BSHII-H121/BSHIII-1E2/BSHIV-MOL361): TCA = (0.6778/0.8401/0.1597/0.5261/0.4015); TCDCA = (0.9252/ < 0.0001/0.6322/0.2703/0.0002); GCA = (0.7814/0.1447/0.6873/0.1938/0.0474); GCDCA = (0.9706/ < 0.0001/0.2851/0.1405/0.9125). BSH nomenclature indicates homolog group (_e.g._ III) and isolate of origin (_e.g._ MOL361). Data are provided as source data file.

measure the individual effects of these bile acid transformations, we expressed _Turicibacter bsh_ genes off of a genomically-integrated high expression vector[48] in the common gut bacterium _Bacteroides thetaiotaomicron_. This bacterium was chosen because it stably colonizes the murine gut and, unlike _E. coli_ C41-pLys, contains a homolog of a characterized 7α-HSDH similar to that of _T. sanguinis_ MOL361[49,50], allowing the engineered bacteria to more completely mimic _Turicibacter_ bile transformations. As this _B. thetaiotaomicron_ strain has previously been described to have some BSH activity[46], we decided to employ the parental strain expressing the non-bile modifying green fluorescent protein (GFP) as a comparison baseline for these experiments (named _Bt_-WT). These engineered _B. thetaiotaomicron_ strains generally transformed tauro- and glyco- conjugated bile acids as predicted based on the BSH characterization in _E. coli_, including preferential TCDCA transformation, indicating they were capable of performing _Turicibacter_-like bile transformations (Fig. 6a, Supplementary Data 2). However, the _B. thetaiotaomicron_ strain that expressed _bsh_ (group III) from strain 1E2 more completely transformed glyco-conjugated bile acids than the strain expressing _bsh_ (group II) from H121, counter to the _E. coli_ findings. This observation led us to not use the _bsh_ (group II)-expressing strain further. Additionally, although we did not notice a significant in vitro growth defect in the engineered _B. thetaiotaomicron_ strains (Supplementary Fig. 6), we observed a bile transformation delay in _B. thetaiotaomicron_ expressing _bsh_ (group I) from MOL361 that we could compensate for by extending the growth period (Fig. 6b).

We monocolonized mice with the _bsh_-expressing _B. thetaiotaomicron_ strains and assessed their circulating lipid profiles and abdominal WAT mass. We found that the engineered strains colonized the gnotobiotic mouse gut at least as well as the control strain, with one strain (BSH-III-1E2) colonizing the cecum slightly better (Supplementary Fig. 7), and that expressing individual _Turicibacter bsh_ genes in the _B. thetaiotaomicron_ background was sufficient to significantly alter host cecal bile levels (Fig. 6c, Supplementary Fig. S8) and the absolute abundance of hundreds of circulating lipid species (Fig. 6d, Supplementary Data 7). In particular, expression of either the group I or group IV _bsh_ led to a decrease in triglycerides (Fig. 6d, e). Expressing the tauro-specific _bsh_ (group IV) from MOL361 also decreased diacylglycerides (Fig. 6g). Expressing the broader specificity _bsh_ (group I) from MOL361 also led to a decrease of phosphatidylglycines, phosphatidylserines, and cholesterol (Fig. 6d–f, h, i). Despite having broad capacity for transformation, _B. thetaiotaomicron_ expressing _bsh_ (group III) from 1E2 did not alter host lipid profiles as much as the other strains (Fig. 6d–i). On a tissue-level, _bsh_-expression also altered WAT storage in the colonized mice, with the broad tauro-deconjugating BSHs (group I and IV) significantly reducing WAT mass (Fig. 6j). Similar to our findings with _Turicibacter_ monocolonization, we also observed a sex difference in BSH responses, with male mice showing more consistent decrease in triglycerides in response to the tauro-specific BSH from MOL361, and females showing more consistent triglyceride responses to the broadly deconjugating BSH (Supplementary Fig. 10). Overall, these BSH-associated lipid and adipose tissues results were consistent with previous findings on MOL361 colonization in the

C57Bl/6 mouse model[38], but differed from MOL361 colonization in our Swiss Webster mice (Fig. 2c, Supplementary Fig. S11).

To further explore potential factors that may drive the cholesterol and WAT alterations in response to _Turicibacter_ colonization and _bsh_ expression, we measured liver transcript levels of farnesoid X receptor, (Fxr), a key nuclear receptor for bile acids; cytochrome P450 Family A Subfamily A Member 1/Cholesterol 7α Hydroxylase (Cyp7a1), the rate-limiting enzyme for conversion of cholesterol into bile acids; and glucose-6-phosphatase (G6pase), a key enzyme for gluconeogenesis. There were no differences in Fxr transcript levels across any of the native and engineered bacterial colonization conditions (Supplementary Fig. 12a). However, we found similar increases in Cyp7a1 expression (Supplementary Fig. 12b) and decreases in G6pase expression (Supplementary Fig. 12c) between the _Turicibacter_ and _bsh_ colonizations. These findings suggest that the bile-modifying activities of _Turicibacter_ lead to downstream alterations in liver expression of bile signaling and response elements. In all, these results demonstrate that expressing strain-specific _bsh_ genes from _Turicibacter_, especially those able to process the abundant taurine-conjugated bile acids present in the murine intestine, is sufficient to alter host cholesterol, bile, and lipid biology.

## Discussion

Results from this study show that several strains of _Turicibacter_ bacteria from the mammalian gut microbiota modulate host bile and lipid compositions. We identified and characterized five novel _Turicibacter_ genes capable of performing bile transformations (four _bsh_, one 7α-HSDH), and revealed that expression of individual _bsh_ genes is sufficient to broadly and differentially alter host lipid profiles. Further, we found that while bile-transforming genes are present in all our surveyed _Turicibacter_ strains, the specific transformation capacity of BSH variants differed by strain in a manner consistent with host environment co-evolution: bile acids in the human gut are a mix of taurine- and glycine-conjugants, whereas murine bile acids are predominantly taurine-conjugants[51,52], providing different bile environments that are preferentially processed by _Turicibacter_ strains isolated from their respective gastrointestinal tracts. This close connection between host-specific bile composition and bacterial modifications may be due in part to the bile sensitivity previously exhibited by MOL361 and/or the high abundance of _Turicibacter_ in the small intestine, causing these bacteria to more strongly associate with host genes for bile reabsorption and lumenal bile levels than other bile-modifying gut bacteria[21,53–57].

This work displays metabolic consequences of colonization by specific gut bacteria and improves the resolution of our understanding connecting specific taxa—in this case, at the strain level—with host physiology. In rodent and human studies, _Turicibacter_ relative abundance often negatively correlates with dietary fats[29,31,58–62] and host adiposity[28,33,34], but some studies have shown opposite relationships[30,63,64]. This could be a result of the phenotypic diversity identified here among _Turicibacter_ isolates wherein the host may experience different lipid outcomes depending on their own specific _Turicibacter_ strains, but could also vary with other features such as

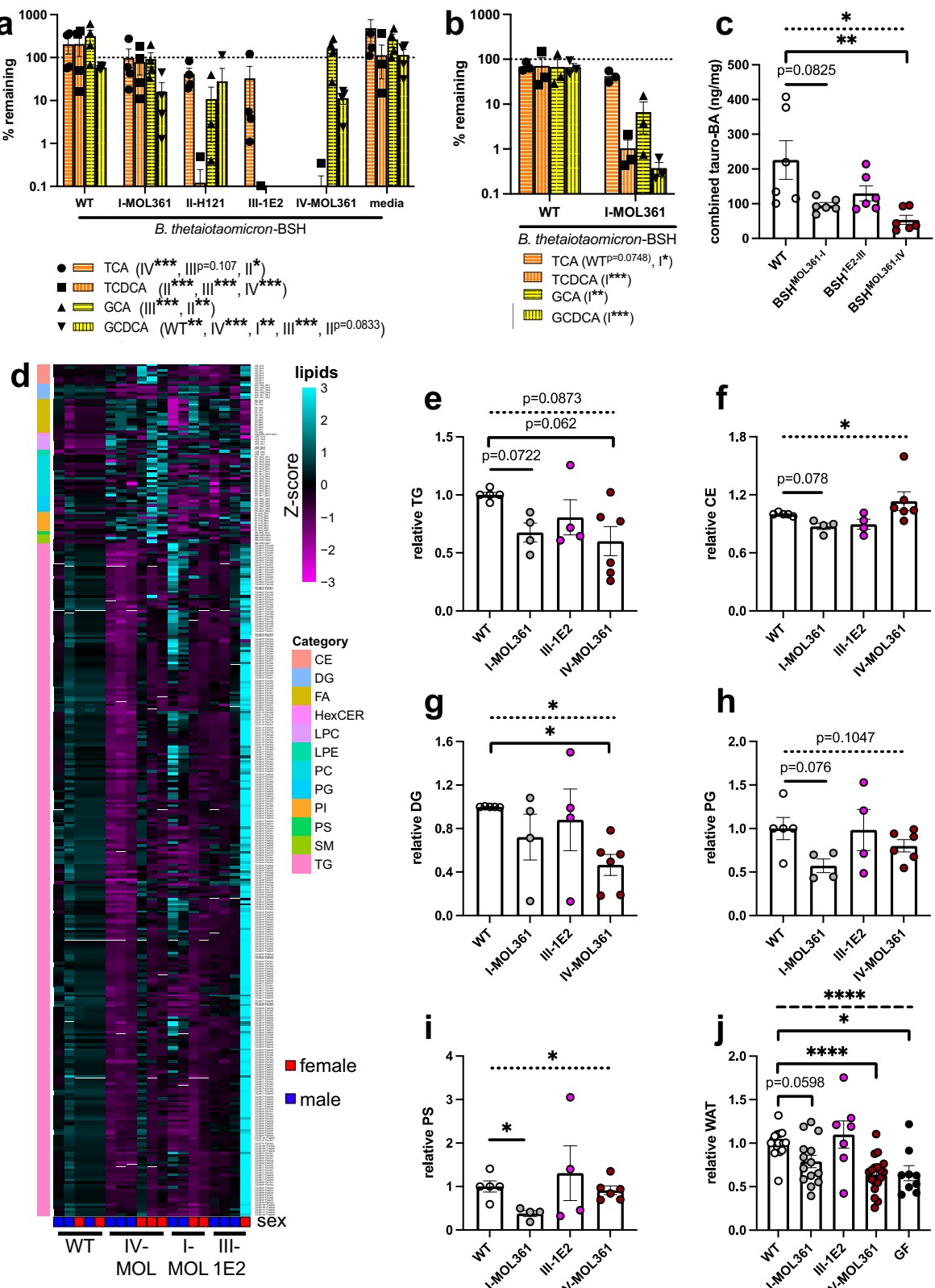

host genetics and sex[21,38] (Supplementary Figs. 3, 9, 10), leading to discrepancies in lipid responses between hosts despite similar bile modifications performed by similar taxa in the gut microbiota. Importantly, effects of *Turicibacter* colonization could also be influenced by biogeographic organization of an individual's microbiota; in addition to the specific taxonomic membership, *Turicibacter* positioning in the small and large intestine may affect host consequences

from their respective bile modifications by transforming bile pools in either section of the gut tract. Despite genomic and localization differences between them, we found that MOL361 and H121 induced lipid metabolite changes that indicate increases in fatty acid oxidation, suggesting that these strains at least share some features that alter host fatty acid metabolism. While we identified broad groups of host lipids that were differentially modulated by different *Turicibacter*

**Fig. 6 | *Turicibacter bsh* expression is sufficient to alter host lipidome and health-associated lipid markers. a** Percent remaining of noted bile acids after 24 h growth with *Bacteroides thetaiotaomicron* expressing noted *bsh* genes. $n = 4$ cultures per strain. **b** Same as **a**, but with 48 h growth with noted *B. thetaiotaomicron* strains. $n = 3$ cultures per strain. For **a** and **b**, points represent individual comparison with media control, legend annotations denote strains with statistical significance for each bile acid using one sample *t*-test comparison with 100% remaining. **c** Quantification of combined taurine-conjugated cecal bile acids (BA) from mice colonized with *bsh*-expressing *B. thetaiotaomicron*. Statistical analysis was performed with Kruskal–Wallis test with Dunn's multiple comparisons test. $n = 6$ animals per colonization. **d** Heatmap of circulating lipid species significantly altered by expression of at least one *Turicibacter bsh* in *B. thetaiotaomicron*. Heatmap Z-score values represent mean abundance of each detected lipid species from labeled lipid categories scaled across all the means of that individual lipid species. Colors on left correspond to lipid class, each column represents one animal. **e–i** Relative combined circulating concentrations of (**e**) triglycerides (TG), **f** cholesterol esters (CE), **g** diacylglycerides (DG), **h** phosphatidylglycines (PG), or **i** phosphatidylserines (PS) of mice monocolonized with *bsh*-expressing *B. thetatiotaomicron*. **j** Relative white adipose tissue weight of mice monocolonized with *bsh*-expressing *B. thetatiotaomicron*. $n$ for each condition in **d–i**: WT = 5(3 M, 2 F);

BSHI-MOL361 = 4(2 M, 2 F); BSHIII-1E2 = 4(3 M, 1 F); BSHIV-MOL361 = 6(3 M, 3 F). $n$ for j: WT = 17; BSHI-MOL361 = 14; BSHIII-1E2 = 7; BSHIV-MOL361 = 18. In (**e–j**), all values normalized to sex-matched littermates, statistical analysis performed by Welch's ANOVA with Dunnett's multiple comparisons to Bt-WT, dotted bar in **c**, **e–j** represents ANOVA statistic for groups below the line. $n$ for (**e–i**): Bt-WT = 5, Bt-BSH-MOL361-I = 4, Bt-BSH-1E2-III = 4, Bt-BSH-MOL361-IV = 6. $n$ for (**j**): Bt-WT = 17, Bt-BSH-MOL361-I = 14, Bt-BSH-1E2-III = 7, Bt-BSH-MOL361-IV = 18. Error bars are mean +/− SEM, *$p < 0.05$, **$p < 0.005$, ***$p < 0.0005$. Figure 6a *p*-value for *B. thetaiotaomicron* expressing BSH in each bile acid (WT/BSHI-MOL361/BSHII-H121/BSHIII-1E2/BSHIV-MOL361): TCA = (0.3020/0.9811/0.0286/0.1074/ < 0.0001); TCDCA = (0.3686/0.6211/ < 0.0001/ < 0.0001/ < 0.0001); GCA = (0.1703/0.8672/0.0028/ < 0.0001/0.3291); GCDCA = (0.0007/0.0035/0.0833/ < 0.0001/ < 0.0001). Figure 6b *p*-values (WT/BSHIV-MOL361): TCA = (0.0748/0.0148); TCDCA = (0.5295/ < 0.0001); GCA = (0.4168/0.0023); GCDCA = (0.1323/ < 0.0001). Figure 6c, **e–j** *p*-values (ANOVA/BSHI-MOL361/BSHIII-1E2/BSHIV-MOL361): 6c = (0.0051/0.0825/ 0.5337/0.0016); 6e = (0.0222/0.0722/0.5767/0.0669); 6f = (0.0377/0.0780/0.2933/ 0.4864); 6g = (0.0142/0.5490/0.9599/0.0072); 6h = (0.1047/0.0760/0.999/ 0.4879); 6i = (0.0076/0.0147/0.9449/0.9344); 6j = (0.0042/0.0457/0.8987/ < 0.0001). Data are provided as source data file.

---

strains, host complexities likely reduced our power to further resolve these differences, pointing to future research that may still reveal more strain-specific effects of *Turicibacter* bacteria.

Further research into *bsh* gene regulation in the *Turicibacter* genus, the relationship between bile and *Turicibacter* colonization and transmission behaviors, as well as the native functionality of the putative BSH and 7α-HSDH homologs we tested, will further explain how these bacteria wield their bile modifications in the intestine. In addition, exploring the discrepencies we observed between bacterially-induced serum and cecal bile profiles could describe other complexities in these effects, such as differential host reabsorption of modified bile acids. Several in vitro bile modulations, such as increases in unconjugated bile acids, were consistent between our in vitro characterizations and *Turicibacter* colonization, but concurrent in vivo increases in expression of bile synthesis genes and levels of glyco-conjugated bile acids seen during *Turicibacter* colonization and expression of individual *Turicibacter* BSH indicate the possibility of more complex interactions between host bile production and *Turicibacter* activity. The fact that the specificity and activity of the individual BSH homologs differed when expressed in different bacterial backgrounds indicates that other unknown cellular or environmental factors influence the way individual BSH act in vivo. This may also include mechanisms that modify the functionality of the other putative BSH homologs we identified from *Turicibacter* that did not deconjugate the specific conjugated bile acids used in our experiments. These findings may influence the ways that bile-modifying genes can be employed to shape host lipid profiles through microbiota engineering, positioning certain microbial species as more appropriate vectors to impart specific host effects. It will also be informative to determine what other activities performed by *Turicibacter* lead to WAT gain in colonized animals, which contrasted findings from colonizing mice with specific *bsh*-overexpressing strains of *B. thetaiotaomicron*. Given this finding, it is likely that *Turicibacter* also influences host lipids through other mechanisms in addition to the bile transformations we characterized.

Our work also connects specific *Turicibacter* members and BSH activity with specific host outcomes. Though some host responses such as broadly decreased triglycerides were consistent across our BSH-recipient mice, the exact lipid and cholesterol responses differed, indicating that the type of deconjugations might have differing connections with host physiology. Further work will continue strengthening the prospect of utilizing *Turicibacter* and/or its bile modifications to intentionally alter host lipid biology to improve host metabolic and lipid-associated health[65,66], as has been proposed with

other bacteria[54,67]. Beyond lipid biology, *Turicibacter* abundance has been positively correlated with diseases such as Parkinson's disease[68] and depression[69], and selective serotonin reuptake inhibitors (SSRIs) have been found to negatively affect *Turicibacter* growth and colonization[70], potentially because they inhibit activity of its unique serotonin transporter[38]. SSRI use is frequently associated with metabolic side effects like weight gain[71,72], and our findings suggests a hypothesis that connects SSRI use and these side effects: SSRI use could diminish gut colonization of bacteria like *Turicibacter*, thus unintentionally altering their impact on host physiology. Future work may develop strategies to reduce interactions between SSRIs and activity of the microbiota, and minimizing the side effects of these drugs and improving host outcomes. In all, these associations further emphasize the importance of understanding mechanisms connecting members of the diverse *Turicibacter* genus to host physiology.

## Methods
### Mouse husbandry
All mouse experiment protocols were approved by the UCLA Institutional Animal Care and Use Committee. Adult (6–8 week old) germ-free Swiss Webster mice were used for all animal experiments (see details in Supplementary Data S3). Mice were reared in flexible gnotobiotic isolators on a 12 h:12 h light dark schedule on standard chow (Labdiet 5K52, 22.1%: 16.6%: 61.3% protein: fat: carbohydrate by calories), then were exited to autoclaved filter top cages with autoclaved chow (Labdiet 5010, 28.7%: 13.1%: 58.2% protein: fat: carbohydrate by calories) and water. For entirety of experiments, mice were kept in 12 h:12 h light:dark cycle under temperature (22°–25 °C) and humidity control. After 1 day of cage acclimation, the noted *Turicibacter* or *Bacteroides thetaiotaomicron* strain was grown in YCFA medium (see below) overnight, pelleted by centrifugation, and resuspended in 1X PBS. Mice were colonized by a 200 μL gavage containing ~$10^6$ colony-forming units (CFU) of *Turicibacter* or ~$10^8$ CFU of *B. thetaiotaomicron*. Alternatively, mice were gavaged with the same volume of PBS alone (referred to as germ-free [GF]) or PBS-suspended fecal slurry from a specific pathogen-free adult mouse (referred to as conventionalized [CONV]). Colonization was measured using strain-specific TuriSERT primers (Supplementary Data 8) and quantitative PCR (qPCR) from weight-normalized contents from the distal small intestine and proximal colon after addition of Low Abundance Microbiota Standard (Zymo) and extraction using the ZymoBIOMICS DNA Miniprep kit (Zymo), referenced to a standard curve of CFU derived from serial dilution plating of cultures of that strain.

## Bacterial culturing

*Turicibacter* isolates and *Bacteroides thetaiotaomicron* strains (Supplementary Data 4) were cultured in a flexible vinyl chamber (Coy) in an anaerobic 85%/10%/5% nitrogen/carbon dioxide/hydrogen mixture (Airgas). *Turicibacter* was grown on Schaedler's agar (BD Biosciences) or modified YCFA[73] (pH 7.4, per liter: 100 mM MOPS, 10 g casitone, 2.5 g yeast extract, 2 g glucose, 2 g maltose monohydrate, 2 g cellobiose, 44 mg $MgSO_4$, 68 mg $CaCl_2$, 0.9 g NaCl, 10 mg hemin, 0.45 g $K_2HPO_4$, 0.45 g $KH_2PO_4$, 4 g $NaHCO_3$, 1 g cysteine, 1 mg resazurin, 1.9 mL glacial acetic acid, 0.7 mL propionic acid, 90 µL isobutyric acid, 100 uL isovaleric acid, 100uL valeric acid, 10 mL ATCC vitamin mixture, 0.2% Tween-80) at 37 °C. Cells were normally grown without shaking, but when appropriate, *Turicibacter* cultures were anaerobically transferred to sealed Hungate tubes or 1.7 mL microcentrifuge tubes and shaken at 225 RPM at 37°C.

For *B. thetaiotaomicron* growth curves, overnight cultures were grown anaerobically in BHI-S for ~48 h at 37 °C to ensure culture saturation, then were subcultured 1:50 for 6 h at 37 °C (final $OD_{600} = 0.41–0.51$). Subcultures were all then diluted to $OD_{600} = 0.1$, and then six replicates were further diluted 1:10 in 100 µL BHI-S in a 96 well plate. Plates were anaerobically sealed with parafilm and incubated at 37 °C. $OD_{600}$ readings were taken every 15 min in a Biotek Synergy H1 microplate reader (Agilent).

*Escherichia coli* C41-pLys (Lucigen) was used for characterizing putative bile modification genes, which were expressed off the pET21+ plasmid. *E. coli* was grown aerobically shaking at 37 °C in Luria Broth (LB, 1% NaCl, 1% tryptone, 0.5% yeast extract) supplemented with 100 µg mL$^{-1}$ ampicillin. Expression of genes was induced by addition of 100 µM IPTG.

## Bacterial isolation and identification

A frozen stool sample from an adult human was thawed on ice and diluted 1:10 with PRAS anaerobic dilution blank medium (Anaerobe Systems). 100 µL of the diluted stool was further diluted to 1:1000 with modified YCFA media containing 0.05% bovine bile, 0.2% Tween-80, and 50 mM resorufin and loaded on Prospector® system arrays (Isolation Bio, San Carlos, CA, USA) following manufacturer's instructions. The fluorescent green signal of the arrays at time 0 was read on the Prospector® instrument in a Coy anaerobic chamber and the arrays incubated at 37 °C in an Anaerobic Systems AS-580 anaerobic chamber (Anaerobe Systems). At 17 and 41 h of incubation the arrays were scanned again and the decrease in green fluorescence from time 0 was used as an indicator for bacterial growth in the array nanowells. Bacteria from the array were transferred to 96-well transfer plates containing 200 µL per well of modified YCFA media, without the addition of 50 mM resorufin. The transfer plates were sealed with a gas permeable film and incubated at 37 °C in a Mitsubishi AnaeroPack jar with a gas-generating sachet (Remel) for 7 days. After incubation, the contents of 538 wells from the transfer plates with visible turbidity were consolidated into secondary 96-well plates, preserved with reduced glycerol, and stored at −80 °C until needed. Unless stated otherwise, all stool and isolate manipulations were conducted anaerobically with a 5% $CO_2$/5% $H_2$/90% $N_2$ atmosphere.

Genomic DNA was extracted in a 96-well format from the consolidated Prospector® culture plates using the Extract All Kit (Applied Biosystems). 20 µL of culture was combined with 20 µL of Lysis Solution and incubated for 10 min at 95 °C, followed by 3 min at room temperature. The DNA was stabilized with the addition of 20 µL of DNA Stabilizing Solution and the resulting DNA lysate stored at −20 °C until needed.

## qPCR screening of novel *Turicibacter* isolates

Genomic DNA from 538 isolates was screened for *Turicibacter* 16S and the *Turicibacter* TuriSERT[38] gene using a multiplexed primer set (Supplementary Data 8). Each 25 µL qPCR reaction mixture had 1 µL Extract All lysate, 10 µL SYBR Power master mix (Applied Biosystems), 0.5 µL of each of the 10 µM primers, and 12 µL molecular grade water. The reactions were run in a QuantStudio 6 Flex (Thermo Fisher) with a 95 °C hold followed by 40 cycles of 95 °C for 15 s, 50 °C for 30 s, 72 °C for 30 s. *Turicibacter sanguinis* MOL361 gDNA and water were used as positive and negative controls, respectively.

## Molecular cloning

*Turicibacter* genes were amplified from template culture lysates with Phusion or Q5 DNA polymerase (NEB) and primers designed to amplify denoted *Turicibacter* genes. pET21- or pWW3837[48]-derived expression plasmids were assembled using Gibson assembly (see Supplementary Data 8 for oligos) for expression in *E. coli* or *Bacteroides thetaiotaomicron*, respectively. Cloned constructs were confirmed through Sanger sequencing prior to functional characterization. pWW3837-derived constructs were cloned into *B. thetaiotaomicron* VPI-5482 as previously described[48,74]. *bsh*-expressing *B. thetaiotaomicron* was compared to *B. thetaiotaomicron* containing the original pWW3837 construct (referred to as wild-type *B. thetaiotaomicron*).

## Genome assemblies

Each strain was streaked on Schaedler agar plates and incubated anaerobically, then an individual colony from each isolate was picked into YCFA medium and grown overnight at 37 °C. DNA was extracted using the ZymoBIOMICS DNA Miniprep kit (Zymo), with bead beating used to lyse cells. Purified genomic DNA was sequenced by MiGS (migscenter.org), and 151 bp paired-end sequences were assembled using CLC Genomics Workbench (Qiagen). Genome assemblies have been deposited at NCBI at BioProject PRJNA846348.

## Whole genome and gene comparisons

anvi'o[75] was used to profile and visualize the different *Turicibacter* strain DNA sequences to locate putative bile salt hydrolase and 7α-HSDH homologs in contig groups, generate variability profiles, and measure gene coverage and detection statistics. Average nucleotide identity (ANI) was calculated using OrthoANIu[76] (available https://www.ezbiocloud.net/tools/ani).

Sequences comparisons between 16S rRNA and *bsh* genes/BSH amino acid sequences were performed in CLC Genomics Workbench (Qiagen). 7α-HSDH sequence comparisons were performed using tblastn (v.20.12.0)[77] using the translated amino acid sequence from *Clostridium absonum*[44].

## Assessment of bile transformations

In vitro characterization of bile transformations by engineered *E. coli* or *B. thetaiotaomicron* strains were performed by growing cells in respective media conditions described above supplemented with 0.5 mM (total combined concentration) of the noted bile species. Cells were grown to stationary phase (shaking at 37 °C), then frozen at −80 °C until further processing. Cells were then thawed, pelleted (5 min at 16,000 × g), and the supernatant was removed to a new microcentrifuge tube. Three volumes of methanol was added, then the mixture was vigorously mixed for 30–60 s and incubated (room temperature, 15 min). Mixtures were centrifuged (5 min, 16,000 × g), the supernatant removed to a clean microcentrifuge tube and dried in a vacuum concentrator. The dried residue was treated with methanol/water/formic acid (50/50/0.1, all by volume) then vigorously mixed and centrifuged as described above. Supernatants were transferred to polypropylene HPLC vials, capped, and maintained at 4 °C while aliquots (typically 5 µL) were injected onto a reversed phase HPLC column (Cadenza CD-C18, 3.0 µm, 250 × 2 mm, Imtakt) equilibrated in solution A (water/formic acid, 100/0.1, vol./vol.) and eluted (0.2 mL minute$^{-1}$) with an increasing concentration of solution B (acetonitrile/formic acid, 100/0.1, vol./vol.); minute/% B: 0/30, 45/70, 48/100, 50/30, 67/30). The effluent from the column was passed

through an electrospray ion source (capillary voltage 42 V, capillary temperature 275 °C, sheath gas flow 15 L min$^{-1}$, spray voltage 5 kV, and −15 kV conversion dynode with −1.2 kV multipliers) connected to a linear ion trap mass spectrometer (Thermo LTQ) scanning from m/z 95-1000 in the positive ion mode. Spectra were recorded and analyzed with instrument manufacturer supplied software. Confirmation of proposed elemental compositions was achieved using the same chromatography and ion source configuration with the spectra recorded by scanning on an orbitrap mass spectrometer (Thermo LTQ XL).

For bile species quantification, an internal spike-in standard of 100 mM chenodeoxycholic acid-D4 (CDCA-D4, Sigma) was added to initial culture supernatants as a normalization reference. Area under the curve from reconstructed ion chromatograms was used to quantify the abundances of each species.

### Serum metabolite analysis

Mice were euthanized with isoflurane and whole blood was collected via cardiac puncture. Blood was allowed to clot in SST Vacutainer tubes (BD) on ice, then centrifuged (4 °C, 1 min, 1500 × g). The supernatant was removed and snap frozen in liquid nitrogen. Serum metabolites were analyzed using global metabolomics platform by Metabolon (Morrisville, NC, USA). Unless otherwise noted, values presented are in arbitrary units (a.u.) for that particular metabolite, determined by the log-transform of the volume-corrected quantification.

### Circulating lipid analysis

Mice were fasted for 4–6 h, then euthanized as described above. Blood was collected via cardiac puncture and deposited into anticoagulatory K$_2$EDTA Vacutainer tubes (BD) on ice. Blood was centrifuged (4 °C, 15 min, 2000 × g), then plasma was collected from supernatant and snap frozen in liquid nitrogen. Shotgun lipidomics was performed by the UCLA Lipidomics Core (Los Angeles, CA, USA) with the following protocol. Thawed plasma was pipetted into glass tubes, a mixture of 70 internal standard lipids (Sciex and Avanti) was added, and lipids were extracted using a modified Bligh and Dyer extraction[78]. The pooled organic layers from two extractions were dried in a vacuum concentrator and resuspended in 50/50 (vol./vol.) methanol/dichloromethane plus 10 mM ammonium acetate. After transfer to robovials, samples were analyzed with a Sciex 5500 with DMS Device (Lipidyzer Platform) using a targeted acquisition list of 1450 lipid species. The Lipidyzer Differential Mobility device was tuned using the EquiSPLASH LIPIDOMIX standard mixture (Avanti). Data were analyzed using an in-house platform using previously described parameters[79], and quantitative values were normalized to input volume. Statistical significance identification for species to include in heatmap for Fig. 6d was performed with uncorrected two-tailed Welch's $t$-test comparison to Bt-WT ($p$-value cutoff <0.05). Both raw and corrected $p$-values are included in Supplementary Data 7.

### Total cecal bile analysis

Mice were colonized as described above and contents from the blind end of the cecum were snap frozen upon sacrifice. Samples were thawed on ice and sent to Metabolon for analysis with total bile analysis pipeline. Values shown at "0" were non-quantified and below limit of detection.

### Adipocyte area calculation

After sacrifice, mice epididimal or gonadal white adipose tissue (e/g WAT) pads were weighed and placed into 4% paraformaldehyde in 1X PBS for 48 h at 4 °C. Fat pads were washed twice in 70% ethanol, then submitted to the UCLA Translational Pathology Core Laboratory (Los Angeles, CA, USA) for paraffin embedding, sectioning, and H&E staining. Ten adipocyte images from each animal (five from each fat pad) were visualized with a 20X objective on an EVOS microscope (Thermo). Adipocyte area for all cells contained entirely within the field of view was automatically measured using the Fiji[80] Adiposoft[81] plug-in (version 1.1.16).

### qRT-PCR measurement of liver transcripts

Gnotobiotic mice were colonized as described above, and upon sacrifice, the median lobe of the liver was dissected and either directly snap frozen in liquid nitrogen (all *Turicibacter* colonized animals) or placed in Trizol, bead beat for 1 min, then frozen in liquid nitrogen (all *B. thetaiotaomicron* colonized animals). All livers were then transferred to −80 °C until further processing. Directly snap frozen livers were thawed overnight at −20 °C in RNALater-ICE (ThermoFisher), then bead beat in Trizol for 1 min, after which all samples were processed in the same manner. RNA was extracted from thawed Trizol samples using the Direct-Zol RNA Miniprep Kit (Zymo), then cDNA was generated using the qScript cDNA Synthesis Kit (Quantabio). qPCR was performed using the PowerUp SYBR Green Master Mix (ThermoFisher) on a QuantStudio5 Real-Time PCR System (ThermoFisher) (primers[82,83] available in Supplementary Data 8) (cycling conditions: 50 °C for 2 min, 95 °C for 2 min, 50 cycles of 95 °C for 15 s, 55 °C for 15 s, 72 °C for 1 min, followed by melt curve). Fold changes in comparison with sex-matched controls (GF for *Turicibacter* colonizations, *Bt*-WT for *B. thetaiotaomicron* colonizations) were calculated using the ΔΔCt method with auto-thresholded Ct values with *ppia* as the housekeeping gene.

### Statistical analysis and reproducibility

Samples sizes were determined using comparison to prior experience, but no statistical method was used to determine sample size. Statistical calculations were performed in in Prism 9.3.1 (Graphpad) or Microsoft Excel 14.7.1. For cecal bile quantifications after *Turicibacter* colonization, outliers were removed using the ROUT method with Q = 1% and removed from analysis, but original data are included in source data files. No other data in this work were excluded from analysis. Dotted lines denote statistical comparison between groups directly below line, solid lines denote pairwise comparison. Unless otherwise noted, all tests were two-sided, $***p < 0.0005$, $**p < 0.005$, $*p < 0.05$, written $p = 0.05 < 0.15$. Heatmaps were created using the *pheatmap*[84] package in R (version 3.6.3)[85].

All data are representative of and/or include biological replicates performed across at least two separate experiments performed on different days. All mouse experiments include samples from both sexes of multiple litters except CONV which included both sexes from one litter. Mice were randomly divided into conditions and tagged by a blinded researcher. Adipocyte image analysis was performed by two blinded researchers. All mass spectrometry and chromatograms are representative of at least two experiments from different days.

### Reporting summary

Further information on research design is available in the Nature Portfolio Reporting Summary linked to this article.

## Data availability

The metabolomics and lipidomics data generated in this study are attached to this manuscript as supplementary data tables. The mass spectrometry datasets are deposited to Metabolights (www.ebi.ac.uk/metabolights/MTBLS7921/) as Project MTBLS7921. Raw sequencing reads and genome assemblies for *Turicibacter* strains are deposited at NCBI under BioProject ID PRJNA846348. The data used to generate the figures are included in source files or supplementary data tables unless provided in the sources mentioned above. Source data are provided with this paper.

## Materials availability

Bacterial strains from this work may be obtained through reasonable request to the corresponding author. Certain materials may be restricted by MTA requirements.

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

## Acknowledgements

*Turicibacter* isolates H121, TA25, T46, and T129 were a gift from Thomas Auchtung. 18F6 and 1E2 were a gift from Kenya Honda. pWW3837 and *B. thetaiotaomicron* VPI-5482 were gifts from Fatima Enam and Justin

Sonnenburg. We would like to thank Alexander Yoon, Gazmend Elezi, and Lauren Seaman for technical assistance with the LC-MS operation and protocols, and Nathan Hwangbo for statistical guidance. This work was supported by funding from the Packard Foundation (E.Y.H.) and Ford Foundation (J.B.L.). E.Y.H. is a New York Stem Cell Foundation—Robertson Investigator. This research was supported in part by The New York Stem Cell Foundation.

## Author contributions

J.B.L., E.L.G, K.C., K.B.Y., J.P., and E.Y.H. conceptualized and planned bacterial and mouse experiments. J.B.L. and K.F.F. designed and performed LC-MS experiments. J.B.L., T.J., A.A., and J.L. performed isolation and characterization of novel Turicibacter isolates. J.B.L. and E.Y.H. wrote manuscript. All authors read and edited final manuscript.

## Competing interests

Findings regarding the host effects of *Turicibacter* reported in the manuscript are the subject of provisional patent application 63/288980, owned by UCLA, on which J.B.L., E.L.G., K.C., and E.Y.H. are inventors. J.B.L. and E.Y.H. are scientific advisors for PurposeBio, which is licensing this patent but which had no role in study design, data collection, data analysis, or interpretation of this work. No other authors have competing interests.
