## [Peer Review File · Nature Communications]

REVIEWERS' COMMENTS

Reviewer #1 (Remarks to the Author):

The authors have addressed the comments of Reviewer 2 by clarifying the statistical analyses used and the incorporation of multiple corrections, and de-emphasized the text supporting strain-specific effects on lipid differences. Only two small questions remain from this set of comments, in regard to the presentation of the statistical analyses.

In response to: identification of lipids to include were done with t-test to Bt-WT (we have added this information at lines 567-568). It would be helpful for the reader to have this information included also in the figure caption of Fig. 2.

In the Fig. 6 caption (line 911), it is written "All values in e-j normalized to sex-matched littermates, n=4-6 animals per colonization, statistical analysis performed by Welch's ANOVA with Dunnett's multiple comparisons to GF." There is however no GF data shown for e-i. If this is really a comparison to data not shown in the figure, this data should be included in the e-i, otherwise it seems to be a typo?

Reviewer #3 (Remarks to the Author):

While I still think this is interesting work, my main concerns remain as in the previous review cycle as no additional data supporting claims was provided.

Minor

Fig. 2b "Serum cholesterol concentrations of mice colonized by noted Turicibacter strains". Figure does not show concentrations, it shows arbitrary units. Same for Fig.3 (i.e., arbitrary unit is not same as concentration)

May 18, 2023 Response to reviewers for Lynch et al.

REVIEWERS' COMMENTS

Reviewer #1 (Remarks to the Author):

The authors have addressed the comments of Reviewer 2 by clarifying the statistical analyses used and the incorporation of multiple corrections, and de-emphasized the text supporting strain-specific effects on lipid differences. Only two small questions remain from this set of comments, in regard to the presentation of the statistical analyses.

Thank you for your evaluation of our updated analyses.

In response to: identification of lipids to include were done with t-test to Bt-WT (we have added this information at lines 567-568). It would be helpful for the reader to have this information included also in the figure caption of Fig. 2.

The pre-identification of lipid species to include in heatmaps was only used for the lipidomics analysis in Fig. 6d, not the metabolomics analysis that included lipids in Fig. 2a. We have clarified this in the methods at line 536 to clarify and connect to the significance screening mentioned in the legend for Fig. 6d.

In the Fig. 6 caption (line 911), it is written “All values in e-j normalized to sex-matched littermates, n=4-6 animals per colonization, statistical analysis performed by Welch’s ANOVA with Dunnett’s multiple comparisons to GF.” There is however no GF data shown for e-i. If this is really a comparison to data not shown in the figure, this data should be included in the e-i, otherwise it seems to be a typo?

Thank you for catching this error. These comparisons were actually made to *Bt*-WT, not GF. We had corrected this at line 898.

Reviewer #3 (Remarks to the Author):

While I still think this is interesting work, my main concerns remain as in the previous review cycle as no additional data supporting claims was provided.

We thank you for your interest in our work. We believe that connecting specific *Turicibacter* bile transformations with specific changes in host physiology is an exciting topic, and we look forward to exploring this in future work.

Minor

Fig. 2b “Serum cholesterol concentrations of mice colonized by noted *Turicibacter* strains”. Figure does not show concentrations, it shows arbitrary units. Same for Fig.3 (i.e., arbitrary unit is not same as concentration)

We have changed “concentration” to “abundance” at lines 834 and 843.